# Risk Factors for Norovirus Infections and Their Association with Childhood Growth: Findings from a Multi-Country Birth Cohort Study

**DOI:** 10.3390/v14030647

**Published:** 2022-03-21

**Authors:** Parag Palit, Rina Das, Md. Ahshanul Haque, Md. Mehedi Hasan, Zannatun Noor, Mustafa Mahfuz, Abu Syed Golam Faruque, Tahmeed Ahmed

**Affiliations:** 1Nutrition and Clinical Services Division, International Centre for Diarrhoeal Disease Research, Bangladesh (icddr,b), Dhaka 1212, Bangladesh; parag.palit@icddrb.org (P.P.); rina.das@icddrb.org (R.D.); md.hasan@icddrb.org (M.M.H.); mustafa@icddrb.org (M.M.); gfaruque@icddrb.org (A.S.G.F.); tahmeed@icddrb.org (T.A.); 2Emerging Infections and Parasitology Laboratory, Infectious Disease Division, International Centre for Diarrhoeal Disease Research, Bangladesh (icddr,b), Dhaka 1212, Bangladesh; zannatun@icddrb.org; 3Faculty of Medicine and Life Sciences, University of Tampere, 33100 Tampere, Finland; 4Department of Global Health, University of Washington, Seattle, WA 98195, USA

**Keywords:** norovirus genogroups, asymptomatic infections, childhood growth faltering

## Abstract

The prevalence of norovirus infections in different geographical locations and their attribution to childhood diarrhea is well established. However, there are no reports showing possible relationships of different norovirus genogroups with subsequent childhood malnutrition. In this study, we attempted to establish a potential association between asymptomatic norovirus infections with childhood growth faltering during. Non-diarrheal stools were collected from 1715 children enrolled in locations in a multi-county birth cohort study across eight different geographical locations and were assessed for norovirus genogroup I (GI) and norovirus genogroup II (GII). Asymptomatic norovirus GI infections were negatively associated with monthly length-for-age Z score/LAZ (β = −0.53, 95% CI: −0.73, −0.50) and weight-for-age Z score/WAZ (β = −0.39, 95% CI: −0.49, −0.28), respectively. The burden of asymptomatic norovirus GI infections was negatively associated with LAZ (β = −0.46, 95% CI: −0.67, −0.41) and WAZ (β = −0.66, 95% CI: −0.86, −0.53) at 2 years of age, whilst the burden of asymptomatic norovirus GII infections was negatively associated with WAZ (β = −0.27, 95% CI: −0.45, −0.25) at 2 years of age. Our findings warrant acceleration in attempts to develop vaccines against norovirus GI and norovirus GII, with the aim of minimizing the long-term sequelae on childhood growth.

## 1. Introduction

The first 2 years of life represent a critical window for the long-term development of a child and malnutrition has been reported to exert detrimental consequences on young children [1]. Malnutrition accounts for 45% of all deaths among children under five [2]. Stunting or linear growth faltering is characterized as a length for age Z score (LAZ) < −2 standard deviations from the WHO child growth standards and is the most prevalent form of chronic malnutrition [3]. Conversely, weight for age Z score (WAZ), the WHO-recommended marker for assessment of childhood underweight, whereby WAZ < −2 represents an underweight child [3]. A recent study consisting of a pooled analysis from 62 low-and-middle-income countries (LMICs) showed that the overall prevalence of stunting among under-fives was 29.1% and underweight was 13.7% [4].

Chronic sub-clinical exposure to fecal enteropathogens as well as the development of environmental enteric dysfunction (EED) are attributes of the underlying pathophysiology of childhood malnutrition. Consequently, childhood diarrhea is a notable risk factor for childhood malnutrition [5,6], whereby diarrheal enteropathogens, such as *Campylobacter*, *Shigella* and enterotoxigenic *Escherichia coli*, being associated with subpar gain in height and weight [6,7]. Norovirus is one of the most prevalent diarrheal enteropathogens and accounts for the second leading cause of diarrhea among children under five and subsequent mortality [8,9]. Human noroviruses have a single-stranded RNA-genome and exhibit a high genetic diversity, with at least 49 known genotypes [10]. Predominantly, norovirus genogroup I (GI) and norovirus genogroup II (GII) are responsible for the majority of non-bacterial cases of gastroenteritis among all age groups [11,12].

Norovirus exhibits a fecal–oral mode of transmission and the clinical symptoms are manifested by an incubation period of 12–48 h with recovery within 2–3 days [11]. A myriad of factors have been deemed to be responsible for repeated and chronic infection by the predominant norovirus genogroups. These factors include the high virulence and infectivity of the norovirus, the continual presence of these noroviruses, the elongated time span of shedding of the viral particles from both symptomatic and asymptomatic individuals and the lack of a long lasting immunity [13]. Consequently, asymptomatic carriage of diarrheal enteropathogens and their subsequent burden among young children is associated with poor growth [14]. However, we do not have any reports showing a potential association between the asymptomatic infections by different norovirus genogroups and the subsequent persistence of these infections in early childhood growth. In this study, we aimed to estimate the incidences of norovirus GI and norovirus GII at different geographical locations and establish possible associations of the burden of asymptomatic infections by these norovirus genogroups with linear growth (assessed by LAZ) and ponderal growth (assessed by WAZ) among children in the first 2 years of life.

## 2. Materials and Methods

### 2.1. Study Settings and Ethical Statement

This current study involves the secondary data analysis of the MAL-ED birth cohort study. Due approval for this secondary data analysis was obtained from the Institutional Review Board (IRB) of icddr,b; which comprises of the Research Review Committee and the Ethical Review Committee, The IRB approval was attained on 20 December 2020 and the protocol approval code was PR-20129. Consequently, MAL-ED (Etiology, Risk Factors, and Interactions of Enteric Infections and Malnutrition and the Consequences for Child Health) was a multi-country birth cohort study conducted across eight distinct geographical locations. In brief, 1715 children from the study sites were recruited from the community within 17 days of birth and were followed until 2 years of age. The study was conducted between December 2009 and February 2012. Due approval was attained from the respective institutional review boards and written informed consent was acquired from the parents or caregivers of each child. The detailed methodology of the study, including the study design, has already been published [15] elsewhere and has been summarized in Figure 1.

### 2.2. Collection of Anthropometric, Socio-Demographic and Morbidity Data

Detailed socio-demographic, socio-economic and anthropometric data were collected upon enrolment and at monthly follow-ups and comprehensive accounts of any morbidity and child feeding practices were acquired during twice-weekly household visits. Length-for-age z score (LAZ) and weight-for-age z score (WAZ) were calculated on the basis of the 2006 WHO standards for children [16].

### 2.3. Collection of Biological Specimens and Biochemical Analyses

Monthly non-diarrheal stools were collected from each participant and venous blood was collected at 7, 15 and 24 months. The biological samples collected were subsequently processed following identical standardized protocols and were stored in −80 °C freezers, prior to further analyses. Plasma zinc, popularly considered to be a proxy indicator for assessment of zinc status in children was assessed by flame absorption spectrophotometry (Shimadzu AA-6501S, Tokyo, Japan). Alpha-1-acid glycoprotein (AGP), a systemic inflammation biomarker, was measured by an immunoturbidimetric assay using commercial kits and a chemistry analyzer (Roche, Munich, Germany). Enteric inflammation was assessed by measuring the levels of myeloperoxidase (Alpco, Salem, NH, USA), neopterin (GenWay Biotech, San Diego, CA, USA) and alpha-1-anti-trypsin (Biovendor, Brno, Czech Republic) at 3, 6, 9, 15, and 24 months by using quantitative ELISA. EED score, ranging from 0 to 10, was calculated from the three biomarkers, as described previously [17].

### 2.4. Detection of Enteropathogens

TaqMan Array Cards (TAC), which are customized multiplex real time-PCR platforms involving compartmentalized primer-probe assays, were used for the detection of enteropathogens [14,18]. Primers specific to the gene encoding the ORF1-ORF2 junction in norovirus GI and norovirus GII were used and cases positive for either norovirus GI and/or norovirus GII were assessed by a threshold Ct of 35. In this study, we used TAC to detect the two predominant genogroups of norovirus, namely: norovirus GI and norovirus GII. We further analyzed the presence of other childhood malnutrition associated pathogens, namely: *Campylobacter* sp., ETEC, EAEC, typical EPEC, *Shigella* sp., *Cryptosporidium* sp. and *Giardia* sp., as described previously [19].

### 2.5. Statistical Analysis

Data were summarized by either mean with standard deviation or by median with interquartile range. Poisson regression was used to calculate the incidence rates for norovirus GI and norovirus GII infections, where the number of infections by norovirus GI and norovirus GII was the outcome variable and the log of several follow ups was the offset variable. Poisson regression models were further used to assess the determinants for the monthly detection of norovirus GI and norovirus GII, using statistical methods as described elsewhere [20,21]. Associations between different genogroups of norovirus with linear growth score (LAZ) and ponderal growth score (WAZ) were estimated using generalized estimating equations (GEE), using multivariate analysis as mentioned in other concurrent studies [22].

The burden of norovirus GI and norovirus GII was defined as the number of infections over the total number of follow ups. Associations between the burden of norovirus GI and norovirus GII infection with linear and ponderal growth at 2 years of age were determined by using multivariate linear regression after adjusting for the site and the necessary covariates, as described above. We excluded the data collected from the participants enrolled at the Pakistan site, owing to bias noted in this site. Multicollinearity between the variables assessed through an evaluation of the variance inflation factor (VIF) and no variable producing a VIF > 5 was found in the final models. The strength of association was evaluated by estimating the β-coefficient and its corresponding 95% CI (confidence interval). A *p*-value of <0.05 was considered to be statistically significant. All statistical analyses were performed in STATA 13.0 (Stata Corporation, College Station, TX, USA).

## 3. Results

### 3.1. General Characteristics of the Study Participants and Incidence Rate of Norovirus GI and Norovirus GII

From the 1715 participants, 34,622 non-diarrheal stool samples were collected. The general characteristics of the study participants have been summarized in Table 1.

Figure 2 and Figure 3 illustrate the incidence rates of norovirus GI and norovirus GII infections. The overall incidence rate of norovirus GII infections was higher than that of norovirus GI infections. The highest incidence of tic norovirus GI infection was found in Bangladesh and the lowest was in Brazil. Conversely, the highest incidence of norovirus GII infections was found in Pakistan while the lowest was in Brazil.

### 3.2. Factors Associated with Infections by Norovirus GI and Norovirus GII

The incidence rate of norovirus GI was significantly lower among male children whereas that of norovirus GII was significantly higher among female children (Table 2).

A longer duration of exclusive breastfeeding, increased WAZ at enrolment, as well as adequate WASH practices were significantly associated with reduced incidences of asymptomatic norovirus GI and norovirus GII infections. On the other hand, increased household crowding associated with increased incidences of asymptomatic norovirus GI and norovirus GII infections. Consequently, increased LAZ at enrolment was significantly associated with reduced incidences of asymptomatic infections by norovirus GI infections. In addition, we did not find any involvement of maternal BMI with incidences of norovirus GI or norovirus GII infections.

### 3.3. Association of Infections by Norovirus GI and Norovirus GII with Linear and Ponderal Growth

We found a significant negative association between monthly linear growth and ponderal growth with norovirus GI infections considering all study sites [β-coefficient: −0.53, (95% CI: −0.73, −0.50); *p* < 0.001], as evaluated using the GEE model (Table 3). This trend was found to be consistent across all study sites except for India in the cases of monthly linear growth for South Africa and Tanzania in the case of monthly ponderal growth, respectively. When assessing the site-specific incidences, we found that the strength of the negative association between monthly linear growth and norovirus GI infection was the highest in Brazil, followed by Bangladesh.

On the other hand, infections with norovirus GII had significant negative associations with monthly linear growth in India, Peru, South Africa and Tanzania and with monthly ponderal growth in Bangladesh, India, Nepal and Peru. However, when the all study sites were considered as a whole, the associations between the overall monthly linear and ponderal growth were not statistically significant [β-coefficient: −0.04, (95% CI: −0.12, −0.35); *p* = 0.281].

Table 4 and Table 5 exhibit the association of the burden of asymptomatic norovirus GI and norovirus GII infections with linear growth and ponderal growth at 2 years of age. In the multivariate analysis models, we also assessed the effect sizes of relevant covariates, including the asymptomatic burden of malnutrition associated pathogens as well as the EED score at 24 months.

The burden of asymptomatic norovirus GI infections had significant negative associations with both linear growth and ponderal growth at 2 years of age, across all the study sites. This means that with one unit increase in the burden of asymptomatic norovirus GI infections, linear growth/LAZ would decrease by 0.46 units and ponderal growth/WAZ by 0.66 units. When assessing the association of the burden of norovirus GI with LAZ at 24 months, we report that strength of association of the burden of *Cryptosporidium* and *Giardia* with LAZ at 24 months was comparable to that of norovirus GI. Consequently, when assessing the association of norovirus GI with WAZ, we report that the individual effect sizes of burden of *Campylobacter* and *Giardia* with WAZ at 24 months was comparable to that of norovirus GII. However, in our multivariate models, the effect size of EED score at 24 months was notably smaller than that of the malnutrition associated pathogens.

The burden of asymptomatic norovirus GII infections had significant negative associations with only ponderal growth at 2 years of age across all study sites, thus indicating that with one unit increase in the burden of asymptomatic norovirus GII infections, ponderal growth/WAZ would reduce by 0.27 units. Here, we also report that although the burden of norovirus GII did not have any significant association with LAZ at 24 months, all other malnutrition associated pathogens and the EED score at 24 months had a significant negative association with this outcome variable. On the other hand, the individual effect sizes of the burden of *Cryptosporidium* and *Giardia* with WAZ at 24 months were greater than those of norovirus GII.

## 4. Discussion

Previous studies in LMICs show that the subclinical norovirus infection rates range from 17%–30% in the second year of life [8,23,24]. We have shown that the overall incidence rate of norovirus GII infections per 100 child-months (12.3%) was greater than that of norovirus GI infections (4.1%). This finding corroborates reports from previously published literature from the MAL-ED birth cohort study, which showed that norovirus GII exhibited a higher attribution burden for diarrhea in the first 2 years of life, compared to norovirus GI [8]; thus indicating that the prevalence of norovirus GII is indeed greater than that of norovirus GI. A cross-sectional study conducted among young Brazilian children with community acquired diarrhea also reported lower detection of norovirus GI in comparison to norovirus GII, although was not considered in this study [12]. Moreover, a significant negative association between the viral loads with LAZ and WAZ was found, but this study did not have a longitudinal design and had monthly follow-ups for the assessment of the burden of multiple enteropathogens including norovirus GI and norovirus GII as well as for the subclinical carriage of norovirus [12].

We have used a systematic and well-synchronized data collection method for the assessment of risk factors for infections by norovirus GI and norovirus GII. Our findings show that, although male sex was found to be protective against infections by norovirus GI, it was found to be significantly associated with increased risk of norovirus GII infections. Secretor status is determined by the expression of the enzymes fucosyl transferase-2 (FUT2) and fucosyl transferase-3 (FUT3) which in turn determines the type of Lewis antigens (Lewis A and Lewis B) in secretions such as saliva, breast milk and blood [25]. The expression of these Lewis antigens in secretions are associated with susceptibility to a number of enteropathogens, including norovirus [26]. Henceforth, our finding involving the varying susceptibility of the male sex to the two norovirus genogroups may be explained by the differential inheritance pattern of the Lewis antigens among the male and female study participants. However, we were not able to determine the secretor status of the mothers nor of the study participants evaluated.

One case-control study conducted in Vietnam reported that young age, crowding in households and recent contact with a symptomatic individual were the prominent key risk factors for symptomatic norovirus infections [27]. Similarly, our analysis showed that household overcrowding (as characterized by mothers having more than three living children and more than two people living in a single room) was significantly associated with a higher incidence of infections by norovirus GI and norovirus GII. Moreover, a lack of WASH practices, a lack of improved floor material in households, a lack of improved drinking water and a lack of improved sanitation along with lower maternal education being associated with increased incidences of norovirus infections. Other studies from similar settings also report that protective roles of improved WASH practices were found against sub-clinical infections by several enteropathogens [21,28].

From our study we also report that household ownership of poultry or cattle did not have any association with incidences of norovirus GI and norovirus GII infections. This finding indicates that poultry or domestic cattle may not act as reservoirs for the transmission of norovirus, unlike other enteropathogens, and is consistent with previous findings that have shown that norovirus is not transmitted to human hosts through animal dung or poultry feces [29]. We also report that a longer duration of exclusive breastfeeding had a significantly protective effect against both norovirus GI and norovirus GII infections. Previous studies have shown protective effects of distinct constituent oligosaccharides in breast milk with protection against several enteric pathogens, including norovirus, through certain immunomodulatory activities of these human milk oligosaccharides (HMOs) [30]. Certain HMOs can act as decoys and bind to the Lewis antigens found in circulation or in the protective mucosa of the GI tract, thus impeding the entry of norovirus into the enterocytes through binding to these Lewis antigens [30,31].

From our analysis, we also report that monthly detection of norovirus GI from non-diarrheal stools was significantly associated with decrements in both LAZ and WAZ, thus implying that repeated sub-clinical infections with norovirus GI were indeed associated with persistent linear and ponderal growth faltering. Consequently, the burden of norovirus GI infections had a significant negative association with both LAZ and WAZ at 2 years of age. However, despite greater incidence rates of norovirus GII infections, the burden of norovirus GII infections were found to be negatively associated with only ponderal growth faltering at 2 years of age. Previous studies conducted on the diarrheal epidemiology of norovirus infections reported that acquired immunity was only found among children with norovirus GII infections [32]. Our findings are thus consistent with the findings from the previous study and imply that the clearance of norovirus GI is slower than that of norovirus GII and that immunity against norovirus GI is more transient in comparison to that against norovirus GII. Henceforth, the prospect of an increased burden of norovirus GI infections becomes pertinent in relation to growth faltering at monthly intervals until 2 years of age.

Although previously published studies conducted in similar settings have reported associations of childhood malnutrition with subclinical enteropathogen infections, these studies did not assess the enteropathogen burdens at successive follow-ups over the study period [14]. The effect sizes of the association of infection with other malnutrition associated viruses with the outcome variables of LAZ and WAZ both at monthly intervals and at 2 years of age were also evaluated in these statistical models. The attribution of the EED, an integral underlying cause of childhood malnutrition, was also adjusted in these models. Studies conducted on duodenal biopsies collected from patients with norovirus diarrhea have shown that norovirus infection leads to disruption in the function of the intestinal barrier and a loss of functions of tight junction proteins, ultimately leading to the clinical manifestations of EED [33]. As a result, reduced intestinal barrier function and a decline in the surface area of the villi for absorption of nutrients, in addition to frequent diarrheal episodes, may possibly have resulted in both linear and ponderal growth faltering [34]. Additionally, reports from studies conducted in LMICs have reported poor quality of dietary proteins leading to deficiency in the levels of circulating essential amino acids [35,36,37]. As a consequence, there may have been a potential downregulation of the intracellular master growth regulation signaling pathway, ultimately leading to a reduction in muscle generation and bone growth [35,38]. Nonetheless, such hypotheses require extensively collected dietary data in the case of a relatively large sample size, which was unavailable for our study.

Irrespective of the insightful nature of our findings, there are several limitations involved with this study. Although we have mentioned that norovirus infection may have resulted in intestinal inflammation and enteropathy, which are responsible for childhood malnutrition, we could not establish any definitive causal link between the burden of norovirus GI and norovirus GII infections with childhood growth faltering. Confirmation of the enteropathy by the gold standard of upper gastrointestinal endoscopy was not carried out and so we lack relevant data on molecular or immunological aberrations in the upper GI biopsy specimens of the study participants. Moreover, we di not evaluate the burden of sub-clinical infections by other enteropathogens among the family members, nor was the prevalence of the diarrheal enteropathogens in the surrounding environment assessed. Henceforth, we are unable to construct any structural framework regarding the biological mechanism for the subclinical transmission of norovirus. Additionally, we could not establish a potential temporal association between infections by norovirus GI and norovirus GII with the ultimate study outcomes, which would require meticulously structured longitudinal studies conducted from birth.

## 5. Conclusions

We report a higher overall incidence of norovirus GII infections per 100 child-months compared to that of norovirus GI infections. Male sex was protective against norovirus GI infections but was associated with increased norovirus GII infections. Longer duration of breastfeeding, improved WASH practices and reduced household crowding had a significant protective association with infections by both norovirus GI and norovirus GII. Asymptomatic norovirus GI infections were associated with monthly decrements in both LAZ and WAZ and the burden of norovirus GI infections was also associated with both linear and ponderal growth faltering at 2 years of age. Conversely, the burden of norovirus GII infections was associated with only ponderal growth faltering at 2 years of age. Henceforth, our findings warrant greater vigilance for the identification of norovirus GI and norovirus GII infections in young children with the aim to avert the long-term sequelae of childhood growth faltering associated with the subclinical infections by these norovirus genogroups. Our findings also mandate expedited attempts to develop vaccines that are able to confer protection against norovirus GI and norovirus GII and the associated long-term adverse effects, particularly on early childhood growth.

## Figures and Tables

**Figure 1 viruses-14-00647-f001:**
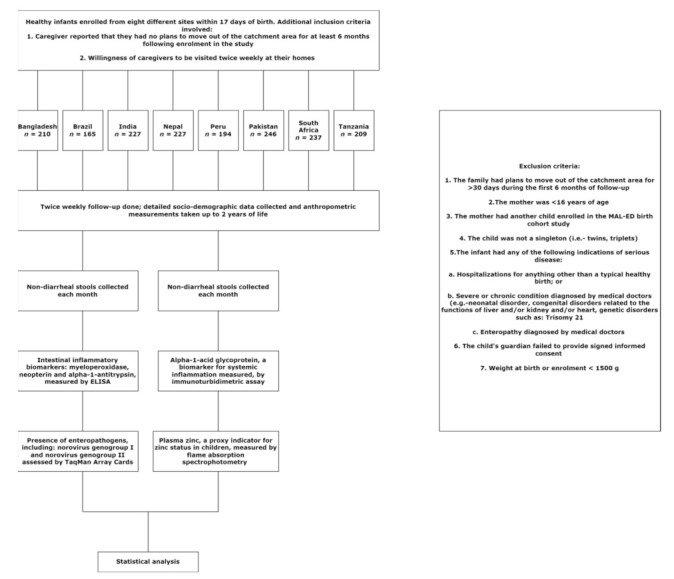
Workflow of the study.

**Figure 2 viruses-14-00647-f002:**
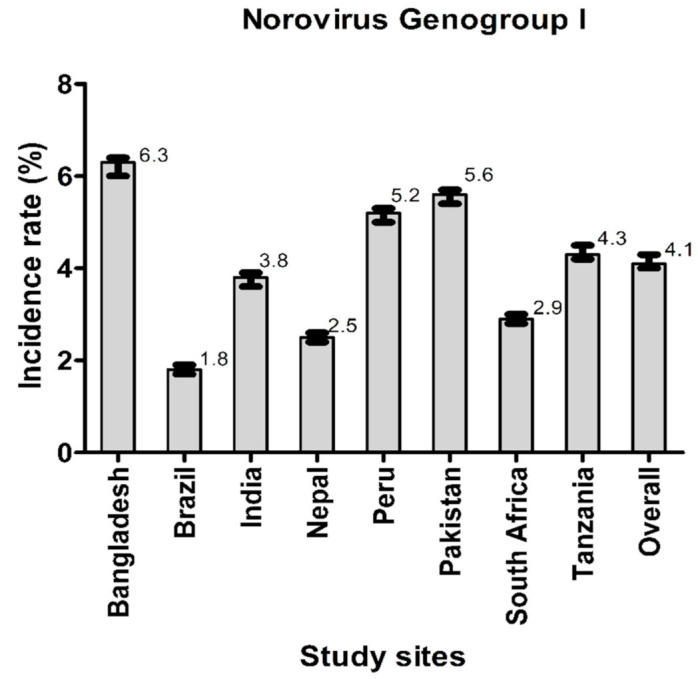
Incidence rates of norovirus GI infections.

**Figure 3 viruses-14-00647-f003:**
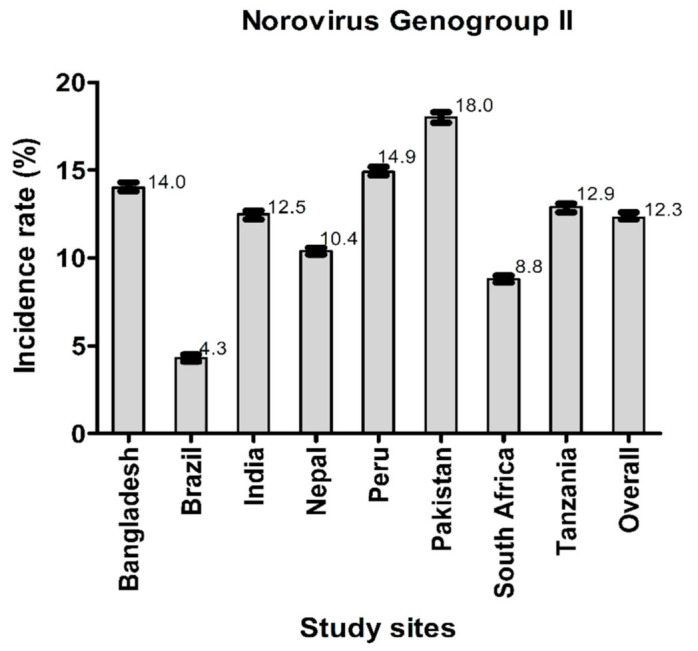
Incidence rates of norovirus GII infections among the participants across all the study sites among all the study participants across all the study sites.

**Table 1 viruses-14-00647-t001:** General characteristics of the study participants enrolled at each of the eight study sites during the study period.

Characteristics	Bangladesh	Brazil	India	Nepal	Peru	Pakistan	South Africa	Tanzania	Overall
Male Sex (*n*, %)	108 (51.4)	89 (53.9)	105 (46.3)	122 (53.7)	105 (54.1)	120 (48.8)	120 (50.6)	105 (50.2)	874 (51.0)
Days of exclusive breastfeeding	143.2 ± 42.7	93.7 ±57.8	105.4 ± 42.9	92.5 ± 54.5	89.5 ± 61.3	19.9 ± 22.7	38.6 ± 26.3	62.2 ± 35	78.6 ± 57.7
Birth weight in kilograms	2.8 ± 0.4	3.4 ± 0.5	2.9 ± 0.4	3.0 ± 0.4	3.1 ± 0.4	2.7 ± 0.4	3.2 ± 0.5	3.2 ± 0.5	3.0 ± 0.5
Weight for age z score at enrollment	−1.3 ± 0.9	−0.2 ± 1.0	−1.3 ± 1.0	−0.9 ± 1.0	−0.6 ±0.9	−1.4 ± 1.0	−0.4 ± 1.0	−0.1 ± 1.1	−0.8 ± 1.1
Length for age z score at enrollment	−1.0 ± 1.0	−0.8 ± 1.1	−1.0 ± 1.1	−0.7 ± 1.0	−0.9 ± 1.0	−1.3 ± 1.1	−0.7 ± 1.0	−1.0 ± 1.1	−0.9 ± 1.1
Length for age z score at 24 months	−2.0 ± 0.9	0.0 ± 1.1	−1.9 ± 1.0	−1.3 ± 0.9	−1.9 ± 0.9	-	−1.7 ± 1.1	−2.7 ± 1.0	−1.7 ± 1.2
Weight for age z score at 24 months	−0.8 ± 0.9	0.5 ± 1.4	−0.9 ± 0.9	−0.3 ± 0.9	0.3 ± 0.9	-	0.5 ± 1.0	0.1 ± 1.0	−0.1 ± 1.1
Maternal age at enrolment (in years)	25 ± 5	25.4 ± 5.6	23.9 ± 4.2	26.6 ± 3.7	24.8 ± 6.3	28.1 ± 5.9	27 ± 7.2	29.1 ± 6.5	26.3 ± 5.9
Maternal BMI at enrolment	22.3 ± 3.4	25.7 ± 4.4	22.0 ± 4.0	25.1 ± 3.2	24.9 ± 3.7	21.5 ± 3.8	27 ± 5.5	22.9 ± 3.2	23.9 ± 4.4
Maternal education level (>6 years of schooling)	77(36.7)	143 (86.7)	147 (64.8)	168(74)	150 (77.3)	44 (17.9)	232(97.9)	134 (64.1)	1095 (63.8)
Mother has more than 3 living children	50 (23.8)	52 (31.5)	70 (30.2)	28 (12.3)	73 (42.8)	141 (57.3)	96 (40.5)	151 (72.2)	671 (39)
Ownership of chicken or cattle (*n*, %)	3 (1.4)	1 (0.6)	14 (6.2)	73 (32.2)	75 (38.7)	144 (62.3)	87 (37.2)	204 (97.6)	601 (35.4)
Routine treatment of drinking water (*n*, %)	130 (61.9)	10 (6.1)	7 (3.1)	98 (43.2)	32 (16.5)	0	12 (5.1)	12 (5.7)	301 (17.6)
Improved drinking water source (*n*, %)	210 (100)	165 (100)	227 (100)	227 (100)	184 (94.9)	246 (100)	196 (82.7)	89 (42.6)	1544 (90.0)
Improved floor (*n*, %)	204 (97.1)	165 (100)	222 (97.8)	109 (48)	69 (35.6)	81 (32.9)	231 (97.5)	13 (6.2)	1094 (63.8)
Improved sanitary latrine (*n*, %)	210 (100)	165 (100)	121 (53.3)	227 (100)	66 (34)	197 (80.1)	232 (97.9)	19 (9.1)	1237 (72.1)
Monthly income less than 150 USD (*n*, %)	69 (32.9)	161 (97.6)	19 (8.4)	106 (46.7)	58 (29.9)	115 (46.8)	179 (75.5)	0	707 (41.2)
Greater than 2 people living per room (*n*, %)	202 (96.2)	24 (14.5)	181 (79.7)	101 (44.5)	72 (31.1)	219 (89.1)	26 (15.2)	114 (54.5)	949 (55.3)
Average serum zinc level (mmol/l) **	11.3 (10.6, 12.1)	14 (13, 14.9)	9.1 (8.6, 9.6)	11.2 (10.4, 12.2)	14.8 (13.1, 17.9)	8.9 (7.7, 10)	22.9 (14.3, 32.9)	11.1 (9.9, 12.3)	11.3 (9.6, 13.7)
Average AGP (mg/dL) ^a^ **	84.3 (71.5, 105)	95.7 (81, 117)	97 (83, 110)	117.7 (102.7, 139)	115 (98, 130)	93 (77.5, 112)	126 (107, 154)	114.3 (97.7, 139)	106.3 (87, 127)

Footnotes: Data represented as mean ± standard deviation, unless otherwise mentioned; ^a^ Average of 7, 15 and 24-months serum zinc and plasma AGP (alpha-1-acid glycoprotein) level; ** median (IQR).

**Table 2 viruses-14-00647-t002:** Factors associated with detection of norovirus GI and norovirus GII in monthly stool samples across each of the eight study sites during the study period.

Risk Factors	Norovirus GI		Norovirus GII	
	IRR (95% CI)	*p*-Value	IRR (95% CI)	*p*-Value
Male sex	0.95 (0.93, 0.97)	<0.001	1.13 (1.11, 1.14)	<0.001
Duration of exclusive breastfeeding	0.95 (0.93, 0.96)	0.030	0.96 (0.96, 0.97)	<0.001
Household ownership of chicken or cattle	0.99 (0.99, 1.01)	0.385	1.00 (0.98, 1.02)	0.832
LAZ at enrolment	0.94 (0.91, 0.95)	0.028	0.99 (0.98, 1.00)	0.197
WAZ at enrolment	0.97 (0.96, 1.01)	0.007	0.98 (0.97, 1.00)	0.005
Maternal age in years	1.00 (0.99, 1.02)	0.008	1.00 (0.99, 1.00)	0.004
Improved drinking water source	0.89 (0.83, 0.95)	<0.001	0.98 (0.95, 1.00)	0.150
Improved floor in households	0.97 (0.95, 0.99)	0.224	0.96 (0.95, 0.99)	0.001
Maternal BMI	1.00 (0.99, 1.03)	<0.001	1.00 (1.00, 1.02)	0.003
Use of water treatment methods	0.85 (0.82, 0.88)	<0.001	0.96 (0.95, 0.97)	0.029
Maternal education more than 6 years of schooling	0.95 (0.92, 0.98)	0.001	0.98 (0.98, 1.01)	0.295
Access to improved sanitation	0.81 (0.79, 0.84)	<0.001	0.96 (0.94, 0.98)	<0.001
Mother has more than 3 living children	1.09 (1.06, 1.13)	<0.001	1.11 (1.08, 1.13)	<0.001
More than 2 people live in per room	1.05 (1.02, 1.08)	<0.001	1.03 (0.99, 1.05)	0.003
Monthly income more than 150 USD	0.92 (0.89, 0.94)	<0.001	0.91 (0.90, 0.94)	<0.001

Footnotes: Poisson regression model was used. Dependent variable was the number of infections during follow up (1–24 months) and offset variable was the log of the total number of follow up. All analyses were adjusted for different study sites and all variables included in the multivariable model.

**Table 3 viruses-14-00647-t003:** Association of infection by norovirus GI and by norovirus GII on linear and ponderal growth of children from enrolment after birth throughout the monthly follow ups until 2 years of age.

Site	Length-for-Age Z Score				Weight-for-Age Z Score			
	Norovirus Genogroup I		Norovirus Genogroup II		Norovirus Genogroup I		Norovirus Genogroup II	
	β-Coefficient (95% CI)	*p*-Value	β-Coefficient (95% CI)	*p*-Value	β-Coefficient (95% CI)	*p*-Value	β-Coefficient (95% CI)	*p*-Value
Bangladesh	−0.36 (−0.38, −0.34)	<0.001	−0.06 (−0.23, 0.05)	0.302	−0.22 (−0.35, −0.15)	<0.001	−0.073 (−0.10, −0.05)	<0.001
Brazil	−0.49 (−0.57, −0.43)	0.01	0.09 (0.07, 0.11)	0.608	−0.30 (−0.433, −0.303)	<0.001	0.086 (−0.11, 0.07)	0.123
India	0.19 (0.09, −0.24)	0.128	−0.12 (−0.24, −0.08)	<0.001	−0.38 (−0.53, −0.33)	<0.001	−0.038 (−0.053, −0.023)	<0.001
Nepal	−0.16 (0.12, 0.23)	<0.001	−0.36 (−0.39, −0.24)	0.223	−0.25 (−0.42, −0.009)	0.003	−0.025 (−0.09, 0.04)	0.03
Peru	−0.18 (−0.32, −0.04)	0.012	−0.15 (−0.32, −0.05)	0.008	−0.36 (−0.52, −0.20)	<0.001	−0.36 (−0.44, −0.25)	<0.001
South Africa	−0.21 (−0.29, −0.10)	0.026	−0.21 (−0.39, −0.17)	0.025	0.02 (−0.02, 0.25)	0.268	0.03 (−0.02, 0.25)	0.835
Tanzania	−0.29 (−0.39, −0.24)	0.003	−0.30 (−0.49, −0.20)	0.003	0.11 (−0.12, 0.35)	0.346	0.11 (−0.12, 0.17)	0.354
Overall	−0.53 (−0.73, −0.50)	<0.001	−0.18 (−0.23, 0.07)	0.671	−0.39 (−0.49, 0.28)	<0.001	−0.04 (−0.12, 0.35)	0.281

Footnotes: Adjusted in Generalizing Estimating Equation model were: sex, age, WAMI Index (water/sanitation, assets, maternal education, and income); enrollment length-for-age z score; maternal BMI; the number of children, poultry/cattle in house, seasonality, serum zinc level, AGP (alpha-1-acid glycoprotein), presence of co-pathogens (*Campylobacter* sp., *Cryptosporidium* sp., *Shigella* sp., ETEC, typical EPEC), site for the overall estimate, and age as the time variable.

**Table 4 viruses-14-00647-t004:** Association of burden of asymptomatic norovirus GI infections with linear growth and ponderal growth at 2 years of age.

Explanatory Variable	Outcome Variables			
	Length-for-Age Z Score at 2 Years of Age		Weight-for-Age Z Score at 2 Years of Age	
	β-Coefficient (95% CI)	*p*-Value	β-Coefficient (95% CI)	*p*-Value
**Burden of asymptomatic norovirus GI infection**	−0.46 (−0.67, −0.41)	0.003	−0.66 (−0.86, −0.53)	<0.001
Male sex	−0.23 (−0.25, −0.20)	<0.001	−0.10 (−0.12, −0.07)	<0.001
Exclusive breast feeding	0.057 (0.05, 0.07)	<0.001	0.07 (0.06, 0.09)	<0.001
Birth weight	0.10 (0.06, 0.14)	<0.001	0.17 (0.13, 0.28)	<0.001
WAMI score	0.98 (0.87, 1.10)	<0.001	0.96 (0.86, 1.07)	<0.001
Maternal BMI	0.30 (0.28, 0.33)	<0.001	0.14 (0.11, 0.16)	<0.001
Concentration of zinc in plasma	0.12 (0.04, 0.15)	<0.001	0.05 (0.01, 0.09)	0.017
Concentration of AGP in plasma	−0.08 (−0.17, −0.04)	<0.001	−0.10 (−0.14, −0.07)	0.005
Burden of asymptomatic *Shigella* infection	−0.57 (−0.71, −0.43)	<0.001	−0.62 (−0.77, −0.43)	<0.001
Burden of asymptomatic *Campylobacter* infection	−0.30 (−0.38, −0.21)	<0.001	−0.63 (−0.79, −0.43)	<0.001
Burden of asymptomatic *Cryptosporidium* infection	−0.45 (−0.64, −0.36)	<0.001	−0.31 (−0.41, −0.28)	<0.001
Burden of asymptomatic *Giardia* infection	−0.48 (−0.55, −0.43)	<0.001	−0.35 (−0.41, −0.28)	<0.001
Burden of asymptomatic ETEC infection	−0.21 (−0.24, 0.10)	<0.001	−0.65 (−0.78, −0.42)	0.02
Burden of asymptomatic typical EPEC infection	−0.15 (−.019, −0.08)	<0.001	−0.11 (−0.14, −0.07)	<0.001
EED score at 24 months	−0.12 (−0.16, 0.06)	<0.001	−0.21 (−0.27, −0.17)	<0.001

**Table 5 viruses-14-00647-t005:** Association of burden of norovirus GII infections with linear growth and ponderal growth at 2 years of age.

Explanatory Variable	Outcome Variables			
	Length-for-Age Z Score at 2 Years of Age		Weight-for-Age Z Score at 2 Years of Age	
	β-Coefficient (95% CI)	*p*-Value	β-Coefficient (95% CI)	*p*-Value
**Burden of asymptomatic norovirus GII infection**	−0.13 (−0.30, −0.04)	0.135	−0.27 (−0.45, −0.25)	0.002
Male sex	−0.24 (−0.29, −0.21)	<0.001	−0.10 (−0.13, −0.08)	<0.001
Exclusive breast feeding	0.06 (0.04, 0.07)	<0.001	−0.07 (−0.08, −0.05)	<0.001
Birth weight	0.27 (0.25, 0.31)	<0.001	0.16 (0.18, 0.15)	<0.001
WAMI score	0.93 (0.84, 1.01)	0.039	0.99 (0.87, 1.03)	<0.001
Maternal BMI	0.06 (0.05, 0.08)	<0.001	0.13 (0.10, 0.16)	0.017
Concentration of zinc in plasma	0.12 (0.09, 0.17)	<0.001	0.05 (0.03, 0.09)	<0.001
Concentration of AGP in plasma	−0.15 (−0.19. −0.07)	0.008	−0.10 (−0.13, −0.09)	0.037
Burden of asymptomatic *Shigella* infection	−0.53 (−0.61, −0.44)	<0.001	−0.64 (−0.77, −0.50)	<0.001
Burden of asymptomatic *Campylobacter* infection	−0.21 (−0.39, −0.18)	<0.001	−0.15 (−0.15, −0.14)	<0.001
Burden of asymptomatic *Cryptosporidium* infection	−0.46 (−0.63, −0.33)	<0.001	−0.33 (−0.51, −0.24)	<0.001
Burden of asymptomatic *Giardia* infection	−0.39 (−0.55, −0.42)	<0.001	−0.34 (−0.40, −0.28)	<0.001
Burden of asymptomatic ETEC infection	−0.14 (−0.18, −0.11)	0.034	−0.18 (−0.26, −0.08)	0.021
Burden of asymptomatic typical EPEC infection	−0.18 (−0.22, −0.15)	<0.001	−0.21 (−0.26, −0.16)	<0.001
EED score at 24 months	−0.13 (−0.17, −0.10)	<0.001	−0.17 (−0.19, −0.11)	<0.001

## Data Availability

A publicly available MAL-ED dataset was analyzed in this study. This data can be obtained from here: ClinEpiDB [https://clinepidb.org/ce/app/record/dataset/DS_841a9f5259], accessed on 3 March 2021.

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
