# Peer review of "Risk Factors for Norovirus Infections and Their Association with Childhood Growth: Findings from a Multi-Country Birth Cohort Study"

_viruses, 2022, doi:10.3390/v14030647_

Round 1

Reviewer 1 Report

Palit et al describe a birth cohort study revealing that asymptomatic infections of noroviruses in <2 years old infants among 8 different countries are associated with their malnutrient. In this study, abundant acquired data were analyzed in detail and has been discussed well. This reviewer agrees that not only symptomatic but also asymptomatic NoV infections influence childhood growth and think that comments pointed below would improve this manuscript before publication.

Major comment

The authors show the result from 1,715 children from 8 countries. Could you provide the period when this data was acquired and discuss what genogroups and genotypes was spread? This information might help readers to consider what genogroups/genotypes influenced on the asymptomatic infection.

Minor comments

Line 48. Genotyping has been changed so this sentence and reference should be updated.

Lines 126-189. Captions and numbers for each section of the results are not accurate.

Lines 176. "NOV" should be "NoV".

Line 230. The sentence "implying that... cattle could act as reservoirs for NoV" does not seem to make sense. Please check here.

Author Response

Thank you for your kind review of our manuscript. Please kindly find the point-by-point responses to your comments and suggestions below.

Palit et al describe a birth cohort study revealing that asymptomatic infections of noroviruses in <2 years old infants among 8 different countries are associated with their malnutrient. In this study, abundant acquired data were analyzed in detail and has been discussed well. This reviewer agrees that not only symptomatic but also asymptomatic NoV infections influence childhood growth and think that comments pointed below would improve this manuscript before publication.

Major comment

The authors show the result from 1,715 children from 8 countries. Could you provide the period when this data was acquired and discuss what genogroups and genotypes was spread? This information might help readers to consider what genogroups/genotypes influenced on the asymptomatic infection.

Response: Thank you for the comment. The study took place between December 2009 to February 2012 and we evaluated the presence of two genogroups of norovirus, namely: norovirus genogroup I and norovirus genogroup II. These are the major norovirus genogroups responsible for enteric diseases in human hosts.

Minor comments

Line 48. Genotyping has been changed so this sentence and reference should be updated.

Response: Thank you for the comment. We have updated the sentence and the reference accordingly.

Lines 126-189. Captions and numbers for each section of the results are not accurate.

Response: Thank you for the comment. We have corrected accordingly.

Lines 176. "NOV" should be "NoV".

Response: Thank you for the comment. In the revised manuscript, we not used the abbreviated form (NoV) and have used “norovirus” instead, as suggested by the second reviewer.

 Line 230. The sentence "implying that... cattle could act as reservoirs for NoV" does not seem to make sense. Please check here.

Response: Thank you for the comment. We have revised the sentence accordingly.

Reviewer 2 Report

Palit et al describe a negative association between LAZ and WAZ with asymptomatic norovirus infections enrolled in the MALED study. What I miss is a norovirus negative comparison group or a group positive for another diarrheal pathogen as comparison. Importantly, as the authors indicate the overall relationships between LAZ/WAZ and norovirus infections are not statistically significant. There is a lot of information in the text and tables that is not relevant for the point the authors are trying to make. Most of the info should be visible from Tables 3-5, but to be frank they are far from user-friendly to interpret (many of the table entries are statistically significant but which ones to focus on is not explained. The authors should highlight the relevant data in the Results section (and not only in the Discussion as in the current draft) (for example “Similarly, our analysis showed that household overcrowding (as characterized by mothers having more than 3 living children and more than 2 people living in a single room) was significantly associated with higher incidence of asymptomatic infections by norovirus GI and norovirus GII” should also be reported in the Results (not only pointing to a Table).

In that way the reader has a better way to follow what data in these tables are supporting their claim.  Showing the differences between the countries of the asymptomatic norovirus rates for GI and GII is nice but is not new and has been reported extensively previously.

I think if the authors just describe the possible risk factors of norovirus infections. I think they could say in the first sentence that they are focusing on asymptomatic infections and in the rest of the manuscript should not repeat ‘asymptomatic’ over and over again.

I think the title “A multi-country birth cohort study reveals crucial associations 2 between asymptomatic infections by different norovirus 3 genogroups with childhood growth” could probably be rephrased to “Risk factors for norovirus infections in a multi-country birth cohort study”.

In the English language a single word such as ‘norovirus’ should not be abbreviated to “NoV’ or ‘NOV’. Please spell out the word.

Author Response

Thank you for your kind review of our manuscript. Please kindly find below the point-by-point responses to your comments and suggestions. 

Point-by-point responses to Reviewer 2

Palit et al describe a negative association between LAZ and WAZ with asymptomatic norovirus infections enrolled in the MALED study. What I miss is a norovirus negative comparison group or a group positive for another diarrheal pathogen as comparison. Importantly, as the authors indicate the overall relationships between LAZ/WAZ and norovirus infections are not statistically significant. There is a lot of information in the text and tables that is not relevant for the point the authors are trying to make. Most of the info should be visible from Tables 3-5, but to be frank they are far from user-friendly to interpret (many of the table entries are statistically significant but which ones to focus on is not explained. The authors should highlight the relevant data in the Results section (and not only in the Discussion as in the current draft) (for example “Similarly, our analysis showed that household overcrowding (as characterized by mothers having more than 3 living children and more than 2 people living in a single room) was significantly associated with higher incidence of asymptomatic infections by norovirus GI and norovirus GII” should also be reported in the Results (not only pointing to a Table). In that way the reader has a better way to follow what data in these tables are supporting their claim.  Showing the differences between the countries of the asymptomatic norovirus rates for GI and GII is nice but is not new and has been reported extensively previously.

Response: Thank you for the comment. Our study is a birth cohort study and not a case-control study. Henceforth, at each follow-up, there were individuals from whom norovirus were detected (cases) and individuals from whom norovirus was not detected (control). However, the cases and the controls differed with each follow-up, being subject to the detection of norovirus. Henceforth, for an overall assessment of the prevalence of the norovirus genogroups, we calculated the incidence rate and the burden of each norovirus genogroup over the entire study period. In our multivariate analysis models, we have adjusted for the diarrheal enteropathogens that are known to be associated with childhood malnutrition, thus accounting for the effect of other diarrheal enteropathogens. As suggested, we have tried to elaborate the results section and have explained the results depicted in the tables. I think if the authors just describe the possible risk factors of norovirus infections.

I think they could say in the first sentence that they are focusing on asymptomatic infections and in the rest of the manuscript should not repeat ‘asymptomatic’ over and over again.

Response: Thank you for the comment. We have revised accordingly.

I think the title “A multi-country birth cohort study reveals crucial associations 2 between asymptomatic infections by different norovirus 3 genogroups with childhood growth” could probably be rephrased to “Risk factors for norovirus infections in a multi-country birth cohort study”.

Response: Thank you for the comment. We have revised the title of our manuscript as per your suggestion.

 In the English language a single word such as ‘norovirus’ should not be abbreviated to “NoV’ or ‘NOV’. Please spell out the word.

Response: Thank you for the comment. We have avoided the use of NoV and have spelt out the word itself, all throughout the manuscript.
